# Effects of 2-Week Exercise Training in Hypobaric Hypoxic Conditions on Exercise Performance and Immune Function in Korean National Cycling Athletes with Disabilities: A Case Report

**DOI:** 10.3390/ijerph17030861

**Published:** 2020-01-30

**Authors:** Hun-Young Park, Won-Sang Jung, Jisu Kim, Hyejung Hwang, Sung-Woo Kim, Younghwan An, Haeman Lee, Seonju Jeon, Kiwon Lim

**Affiliations:** 1Physical Activity and Performance Institute (PAPI), Konkuk University, 120 Neungdong-ro, Gwangjin-gu, Seoul 05029, Korea; parkhy1980@konkuk.ac.kr (H.-Y.P.); jws1197@konkuk.ac.kr (W.-S.J.); kimpro@konkuk.ac.kr (J.K.); hfilm@konkuk.ac.kr (H.H.); kswrha@konkuk.ac.kr (S.-W.K.); 2Department of Sports Medicine and Science, Graduate School, Konkuk University, 120 Neungdong-ro, Gwangjin-gu, Seoul 05029, Korea; 3Korea Paralympic Committee, 424 Olympic-ro, Songpa-gu, Seoul 05540, Korea; ayhayh2@koreanpc.kr (Y.A.); passno6@koreanpc.kr (H.L.); jeonfam@koreanpc.kr (S.J.); 4Department of Physical Education, Konkuk University, 120 Neungdong-ro, Gwangjin-gu, Seoul 05029, Korea

**Keywords:** hypobaric hypoxic training, exercise performance, immune function, athletes with disability

## Abstract

We aimed to evaluate the effects of a 2-week exercise training program in hypobaric hypoxic conditions on exercise performance and immune function in Korean national cycling athletes with disabilities. Six Korean national cycling athletes with disabilities participated in exercise training consisting of continuous aerobic exercise and anaerobic interval exercise in hypobaric hypoxic conditions. The exercise training frequency was 60 min (5 days per week for 2 weeks). Before and after the exercise training, exercise performance and immune function were measured in all athletes. Regarding the exercise performance parameters, the 3-km time trial significantly decreased and blood lactate levels after the 3-km time trial test significantly increased by exercise training in hypobaric hypoxic conditions. Regarding the oxygen-transporting capacity, significant differences were not observed. Regarding immune function, the number of leukocytes and natural killer cells significantly decreased and that of eosinophils, B cells, and T cells significantly increased. These results indicated that our 2-week hypoxic training showed the potential to improve exercise performance in Korean national disabled athletes. However, the effects of our hypoxic training method on immune function remained unclear.

## 1. Introduction

Since the 1968 Mexico City Olympics, various studies have been conducted to assess the performance of exercise training in hypoxic conditions to enhance the performance of athletes. Currently, these training methods are commonly applied by several athletes and coaches [1,2,3,4]. Altitude/hypobaric hypoxic training can be generally classified as follows: (1) living high-training high (LHTH), which was the first design of living and training at 1500–4000 m in naturally high-altitude environments; (2) living high-training low (LHTL), defined as living at 2000–3000 m but training at or near sea level; and (3) living low-training high (LLTH), defined as living at or near sea level but training at 2000–4000 m. Hypoxic training methods mainly aim to improve aerobic exercise performance; however, recently, they have been applied to exercise events and training purposes [5].

Endurance exercise is significantly influenced by the oxygen-transporting capacity of the body to localized areas, such as the active muscles [6]. Enhancing the oxygen-transporting capacity increases the efficiency of aerobic energy production and, consequently, enhances the maximum oxygen uptake (VO_2_max) and exercise performance by improving the time until fatigue and increasing exercise intensity [6,7]. Altitude/hypobaric hypoxic training improves athletic performance in aerobic events by enhancing the oxygen-transporting and oxygen-utilizing capacities through hematological changes, such as those in hemoglobin (Hb) mass and erythropoiesis, and by non-hematological changes, such as those in cardiovascular function and oxygen availability in the skeletal muscles [8,9,10]. 

Recently, a new altitude/hypobaric hypoxic training method, termed repeated sprint training in hypoxia (RSH), has been applied to improve the exercise performance of anaerobic athletes and team sport athletes. RSH is performed by repeating 10 or more high-intensity exercise sessions within 30 s for 4–6 weeks at 2000–4000 m in a simulated hypoxic condition. RSH improves anaerobic exercise performance more effectively with significant changes in molecular levels and blood perfusion in the active muscle than by repeated sprint training in normoxic conditions, subsequently improving lactate tolerance, enhancing the acid-base balance, and increasing the activity of the enzymes involved in glycolysis [4,11,12,13]. However, studies that examined the positive effects of exercise performance by exercise training in hypoxic conditions for national team athletes with disabilities are significantly insufficient. In Korea, only one study evaluated a 2-week altitude/hypobaric hypoxic training in three national cyclists with disabilities before the 2016 Rio Paralympic Games [14].

Moreover, exercise in hypoxic conditions acts as a stressor, significantly changing the individual’s physiological and metabolic functions [15]. Exposure to hypoxic conditions results in changes in the nervous and endocrine systems, subsequently affecting the individual’s immune function [16]. Exposure to hypoxic condition increases the levels of interleukin (IL)-6, which is possibly due to the increased epinephrine released by increased β-adrenergic pathway activity in the early stages of exposure to hypoxic conditions and increased norepinephrine secretion due to increased α-adrenergic pathway activity as a result of long-term exposure to hypoxic conditions [15,16,17,18]. Increased IL-6 levels caused by hypoxic conditions may enhance the oxygen-transporting capacity via the activation of the vascular endothelial growth factor (VEGF) and increased production of erythropoietin (EPO) and reticulocyte [19,20]. The most representative changes in immune function following exposure to hypoxic conditions include decreased CD4+ T-cell counts, decreased T-cell activation and proliferation, lymphocytosis, neutropenia, and inflammatory upregulation of cytokines, such as interleukin(IL)-6, IL-1, C-reactive protein, and tumor necrosis factor-α [16,18,20]. However, studies that examined the changes in immune function following exercise training in hypoxic conditions in national athletes with disabilities have not been conducted. Particularly, it is important to examine the effects of exercise training in hypoxic condition on immune function in disabled individuals, who are more physiologically sensitive to immunological responses than nondisabled individuals.

Therefore, the present study aimed to examine the effects of 2-week exercise training in hypobaric hypoxic conditions, corresponding to a simulated altitude of 2000–3000 m, on exercise performance and immune function of Korean national cycling athletes with disabilities. We hypothesized that a 2-week exercise training program in hypobaric hypoxic conditions would enhance exercise performance and would not adversely affect immune function in Korean national cycling athletes with disabilities.

## 2. Materials and Methods

### 2.1. Subjects

The subjects included South Korean national cycling athletes with disabilities (n = 6) who were preparing for the 2020 Tokyo Paralympics. The characteristics of the athletes are presented in Table 1. The six South Korean national cycling athletes with disabilities included two blind athletes (one male and one female), two blind pilots (one male and one female), one cognitively-impaired male athlete, and one male athlete with a spinal cord disability. There was no significant change in the physical characteristics of the athletes from exercise training in hypoxic conditions. The purpose and methods of the study were comprehensively explained to the subjects. The subjects provided written informed consent after they received sufficient explanation regarding the study and completely understood the possible adverse effects of the study. The trial was registered and disclosed at the institutional review board of Konkuk University (7001355-201812-HR-286) and clinical trial information in the Clinical Research Information Service (KCT0003952) in Korea.

### 2.2. Study Design

The experimental design involved the following: a 2-day pretest session, a 2-week session of hypobaric hypoxic training, and a 2-day post-test session.

All South Korean national cycling athletes with disabilities participated in a 2-day pre- and post-test session. On the first testing day, blood pressure and body composition were measured and blood samples were collected between 8:00 and 10:00 a.m. after a day of fasting for the analysis of blood variables related to oxygen-transporting capacity and immune function at sea level. Subsequently, after the athletes were allowed to eat and rest for 4 h, the graded exercise test (GXT) was performed at normoxic conditions. On the second testing day, the 3-km time trial test was conducted at Gwangmyeong Speedom, an official cycling stadium, at normoxia.

After the pretest session, all athletes performed the following two exercise training in 90-min sessions: warm-up, continuous training, interval training, and cool-down. Exercise training was performed in hypobaric hypoxic conditions (Week 1 (simulated altitude: 2000 m; 596 mmHg), Week 2 (simulated altitude, 3000 m; 526 mmHg)) created by the environmental control chamber (NCTC-1, Nara Controls, Seoul, Korea). The training frequency was 90 min (5 days per week for 2 weeks). Warm-up and cool-down were set at 50% maximal heart rate (HRmax) for each subject for 5 min and subsequently increased by 10% HRmax every 5 min and performed for 15 min. The continuous training sessions consisted of 30-min continuous cycling exercise, corresponding to 80% of the HRmax. The running velocity on a cycle was changed depending on the heart rate monitor (Polar M400, Helsinki, Finland) to match the 80% HRmax. The interval training sessions consisted of 15 repetitions of interval cycling exercise (30-s exercise corresponding to 95%–100% of the HRmax and 90-s rest).

All testing and exercise training procedures were performed in a 9 m (width) × 7 m (length) × 3 m (height) chamber with a temperature of 22 ± 1 °C and a humidity of 50 ± 5%, regulated by the environmental control chamber (NCTC-1, Nara Controls, Seoul, Korea).

All exercise training sessions in hypoxic conditions were supervised by directors, coaches, and researchers.

### 2.3. Blood Pressure

After the athletes sufficiently rested for more than 20 min, their blood pressure was measured twice using an autonomic blood pressure monitor (HBP-9020, Omron, Tokyo, Japan) and the average value was used as a result value. If the results of the first and second measurements were different, the measurement was repeated after 10 min of additional rest.

### 2.4. Body Composition

Weight, fat-free mass, and percent body fat were analyzed as the body composition parameters using a dual-energy X-ray absorptiometry device (Primus, Osteosys, Seoul, Korea). To measure body composition using a dual-energy X-ray absorptiometry device, the subject was measured with the entire body in the center of the examination table, with both feet slightly rotated inward and the shoulders and waist immobile.

### 2.5. Exercise Performance

To assess the aerobic performance, the maximal oxygen uptake (VO_2_max) and 3-km time trial test results were evaluated. First, the graded exercise test was performed before and after training using the METAMAX 3B autonomic respiratory metabolic analyzer (Cortex, NY, USA) and the LC6 Monark ergometer (Monark, Sweden) for the lower limb and the 881E Monark ergometer (Monark, Vansbro, Sweden) for the upper limb. In one athlete with a spinal cord disability, the test started at 70 watts (1 kp and 70 rpm) and increased by 1 kp every 2 min until exhaustion. Other athletes started at 100 watts (1 kp and 50 rpm) and increased by 0.5 kp every 2 min until exhaustion. Respiratory gas data were measured every 10 s during the GXT. The VO_2_max was the highest VO_2_ obtained during the concluding period of the exercise test. The GXT was completed when the following three items were satisfied: (1) VO_2_ plateau—no further increase in oxygen use per minute even with an increase in work performed, (2) The heart rate (HR) within 10 beats of the age-predicted HRmax—this was the basis for using participants’ HRmax as a surrogate for the VO_2_max when designing personal training programs, and (3) plasma (blood) lactate concentrations of >7 mmol/L. The effectiveness of 2-week exercise training in hypobaric hypoxia was evaluated using the measured values of the VO_2_max, maximal minute ventilation (VEmax), HRmax, and blood lactate level. Moreover, the 3-km time trial test was conducted at Gwangmyeong Speedom, an official cycling stadium, and the measured 3-km time trial records and blood lactate levels were used as the measurement values. Blood lactate level was measured in duplicate by fingertip method using an automated blood lactate analyzer (Lactate Pro 2, ARKRAY Inc., Kyoto, Japan).

### 2.6. Oxygen-Transporting Capacity and Immune Function

To assess oxygen-transporting capacity and immune function, the levels of red blood cells (RBC), Hb, hematocrit (Hct), EPO, white blood cells (WBCs), eosinophils, neutrophils, basophils, natural killer (NK) cells, B cells, and T cells were measured before and after hypobaric hypoxic training.

A total of 10-mL of blood samples were collected between 8:00 and 10:00 a.m. after a day of fasting. Then, 4 mL of blood was placed in a heparin-coated tube and centrifuged at 3500 rpm for 10 min and the serum was collected and rapidly frozen at −70 °C. Three milliliters of blood was placed in a tube without an anticoagulant and centrifuged at 3500 rpm for 10 min. Subsequently, the plasma was collected and stored at a rapid freezing rate at -−70 °C. The remaining 3 mL of whole blood was stored in the refrigerator in a heparinized tube. Subsequently, the frozen or refrigerated plasma, serum, and whole blood were commissioned by the Clinical Laboratory of Green Cross Medical Foundation and analyzed using several methods.

In detail, the SYSMEX XN 9000 (Sysmex, Kobe, Japan) hematology analyzer was used for the analysis of RBC, Hb, Hct, WBC, neutrophil, eosinophil, and basophil. RBC and Hct were measured by the electronic impedance method using the Cellpack Kit (Sysmex), Hb was measured using cyanide-free Hb spectrophotometry using the Cellpack Kit (Sysmex), and WBC, neutrophil, eosinophil, and basophil were measured by flow cytometry using the Cellpack Kit (Sysmex). EPO was measured using an Immulite 2000 XPI analyzer (Siemens, Eschborn, Germany) using the chemiluminescent immunoassay. NK cells, B cells, and T cells were analyzed using FC500 (Beckman Counter, CA, USA) and measured by flow cytometry using the NK cell kit (Beckman Coulter, Paris, France), CD19-PE kit (Beckman Coulter, Paris, France), and CD3-PC5 kit (Beckman Coulter, Paris, France), respectively.

### 2.7. Statistical Analysis

Means and standard deviations were calculated for each primary dependent variable. To evaluate the effect of hypobaric hypoxic training in South Korean national cycling athletes with disabilities, we performed a nonparametric statistical method, Wilcoxon’s signed-rank test, considering that the subjects could not be normally distributed to less than 10 subjects. All analyses were performed using the Statistical Package for the Social Sciences version 24.0 (International Business Machines Corp., Armonk, NY, USA). The level of significance was set at 0.05.

## 3. Results

### 3.1. Exercise Performance

The effects of 2-week exercise training in hypobaric hypoxic conditions on the exercise performance parameters are shown in Figure 1. Among the parameters, the 3-km time trial record (*p* = 0.027), which were directly associated with exercise performance, showed a significant improvement with the 2-week exercise training in hypobaric hypoxic conditions. Furthermore, blood lactate levels immediately after the 3-km time trial test significantly increased (*p* = 0.043). However, the VO_2_max did not increase significantly.

### 3.2. Oxygen-Transporting Capacity in the Blood

As shown in Figure 2, there were no significant differences in the oxygen-transporting capacity parameters after the 2-week exercise training in hypobaric hypoxic conditions.

### 3.3. Immune Function

Figure 3 presents the pre- and post-test data for immune function for the 2-week exercise training in hypobaric hypoxic conditions. Among the various immune function parameters, WBC (*p* = 0.028) and NK cell levels (*p* = 0.027) significantly decreased after the exercise training but eosinophil (*p* = 0.028), B-cell (*p* = 0.046), and T-cell (*p* = 0.046) levels significantly increased. Changes in other variables, as well as neutrophil and basophil counts, were not significant.

## 4. Discussion

Our study investigated the effects of a 2-week exercise training in hypobaric hypoxic condition corresponding to a simulated altitude of 2000–3000 on exercise performance and immune function of Korean national cycling athletes with disabilities.

### 4.1. Exercise Performance

In the present study, the 2-week exercise training in hypobaric hypoxic conditions showed potential to improve exercise performance by decreasing the 3-km time trial records in South Korean national cycling athletes with disabilities.

Generally, exercise performance enhancement is influenced by several factors, including the nervous system, hormones, and various proteins, that significantly control the muscle tissue, resulting in efficient oxygen and energy utilization and hematological factors, such as improved oxygen-transporting capacity in the blood [21]. Additionally, the non-hematological factors include increased oxidative enzyme activity [22,23], increased amount and density of the mitochondria [24,25,26,27], increased energy-utilizing capacity and changes in substrate utilization [9,28,29], enhanced blood lactate level tolerance and acid-base balance in the muscles [4,30], improved blood rheological and hemodynamic functions [10,31], enhanced intracellular iron delivery capacity [32], increased autonomic nervous system balance [33,34], changes in various hormone secretions [9,35], and increases in various proteins associated with oxygen utilization [36,37]. The physiological changes in the abovementioned non-hematological factors had positive effects on the 3-km time trial records after our 2-week exercise training in hypobaric hypoxic conditions. Recently, the application rate of living high (e.g., LHTH and LHTL methods), which induces hematological changes, is decreasing, and the LLTH method, which is considered more efficient than the LHTH and LHTL methods, has been frequently used [2,5]. Consistent with the present study, where the absolute value of VO_2_max did not increase significantly, but the relative value showed a significant increase, various studies have demonstrated an improvement in the VO2max or 3-km time trial records by short-term (2- to 4-week period) exercise training in hypobaric hypoxic conditions [1,6,8,14].

The abovementioned non-hematological variables were not analyzed in this study. However, considering that significant changes in hematological variables, such as the oxygen-carrying capacity in the blood (e.g., RBC, Hb, Hct, and EPO), were not observed, it can be hypothesized that the 3-km time trial records of national cycling athletes with disabilities were improved by several non-hematological variables. Additionally, in the present study, considering that we observed higher blood lactate levels after the 2-week exercise training in hypobaric hypoxic conditions immediately after the 3-km time trial test, it can be inferred that short-term hypoxic training improves blood lactate tolerance, defined as the blood lactate’s ability to resist substances causing fatigue. Although the subjects and training periods in hypoxic conditions were different, Park & Lim [38] determined the effectiveness of 6-week intermittent hypoxic training (IHT) at 3000 m hypobaric hypoxic conditions vs. intermittent normoxic training (INT) on aerobic and anaerobic exercise capacity in competitive swimmers. The IHT group showed a significant increase in VO_2_max and blood lactate levels after the exercise test compared with the INT group. They insisted that the IHT method may be effective in the improvement of exercise performance by enhancing blood lactate tolerance in competitive swimmers. Hence, we believe that the 3-km time trial records of the South Korean national cycling athletes with disabilities improved due to increased tolerance to blood lactate levels, and these findings are consistent with the findings in general elite athletes [4,38]. This increased tolerance to fatigue substances is considered beneficial for athletes who must repeatedly participate in high-intensity exercise training and competition.

Additionally, exercise training in hypobaric hypoxic conditions leads to the adaptation of chemoreceptors to a variety of acidic substances, resulting in increased minute ventilation by hypoxic ventilatory response, a response to the release of more acidic substances in the body through respiration [39]. In our study, we believe that VEmax, which was not statistically significant but increased after the 2-week exercise training in hypobaric hypoxic condition, had a significant effect on the enhancement of exercise performance (e.g., VO_2_max or 3-km time trial records).

### 4.2. Oxygen-Transporting Capacity of the Blood

Exposure to altitude/hypobaric hypoxic conditions stimulates erythropoiesis and improves the oxygen-transporting capacity of the blood in individuals residing at sea level [40]. Erythropoiesis stimulated by hypoxic conditions is manifested by an increase in Hb mass and RBC count [40,41]. Furthermore, EPO is the most influential factor in the hematological changes brought about by hypoxic conditions [6]. Decreased oxygen levels in the blood by hypoxic conditions stimulates the release of the glycoprotein hormone EPO in the kidneys, thereby increasing RBC production in the red blood marrow [6,41]. When exposed to altitude/hypobaric hypoxic conditions, EPO acutely increases its secretion and reaches maximum secretion in approximately 48–72 h. The production of new RBCs by EPO stimulation takes approximately 5 days under continuous hypoxic conditions [41,42]. Erythropoiesis caused by exposure to hypoxic condition leads to a decrease in the plasma volume, indicating that the ability to transport oxygen per blood unit is increased. However, a decrease in the plasma volume may adversely affect the oxygen-transporting capacity of the blood. Exercise training in hypoxic conditions is significantly effective in improving the oxygen-transporting capacity of the blood because it increases the plasma volume and RBC production by EPO and this effect persists for approximately 16 days after exposure to the hypoxic condition [32,40].

The hypoxic training applied to the athletes in our study was the LLTH method, defined as living at or near sea level but being exposed to the hypoxic condition only during exercise training. Since the athletes were only substantially exposed to hypobaric hypoxic conditions for less than 2 h per day in the LLTH method, hypoxic stimulation may be insufficiently achieved. Thus, the LLTH method does not result in changes in hematological variables [4,29]. In the present study, the most accurate erythropoiesis evaluation method (e.g., the measurement of RBC count and Hb mass) was not performed, but, as a result of examining various parameters related to the oxygen-transporting capacity of the blood, no significant change was observed in all parameters. These results suggest that hypoxic training using the LLTH method does not enhance exercise performance and does not improve the oxygen-transporting capacity of the blood.

### 4.3. Immune Function

Performing exercise in an altitude/hypobaric hypoxic condition acts as a stressor, significantly changing the individual’s physiological and metabolic functions and nervous and endocrine systems and, consequently, affecting their immune function [15,16]. The following representative changes in immune function are observed when exposed to hypoxic conditions: decreased T-cell numbers, decreased T-cell activation and proliferation, increased neutrophil counts, and upregulation of inflammatory markers, including cytokines such as IL-6 and IL-1, C-reactive protein, and tumor necrosis factor (TNF)-α [43,44,45]. Consistent with the previous studies examining the changes in immune function via hypoxic training, in our study, exercise training in hypoxic conditions significantly decreased the WBC count and immunoglobulin levels [45,46,47].

In the present study, we examined the effects of 2-week exercise training in hypobaric hypoxic conditions on the immune function of South Korean national cycling athletes with disabilities. The normal values of WBC ranged from 4.0 × 10^3^/uL to 10.0 × 10^3^/uL and there were several types of WBCs in the circulating blood. WBCs can be classified into eosinophils, neutrophils, basophils, monocytes, and lymphocytes according to their size and shape by leukocyte differential calculation. Generally, WBC count increases in the case of infections caused by pathogens or tissue damage through external stress stimulation. We found that the WBC count in our athletes decreased after hypoxic training, which is consistent with the results of previous studies [45,47]. However, since the decreased WBC count was within the normal range, we concluded that the hypoxic training applied in our study does not adversely affect the athletes’ immune function. Additionally, we believe that repetitive hypoxic training may increase the WBC function and decrease the number of WBCs within the normal range in cycling athletes with disabilities.

A significant increase in eosinophil counts was observed in this study, which was possibly due to the increased activation of VEGF, IL, and TNF-α when exposed to hypoxic conditions [48,49,50]. However, since the levels of eosinophil, neutrophil, and basophil were within the normal ranges before and after the hypobaric hypoxic training, a significant change in the athletes’ immune function was not observed.

Lymphocytes (e.g., NK cells, B cells, and T cells) play a significant role in immune function. NK cells are responsible for the innate immune function and maturation of the liver or bone marrow and attack virus-infected cells or tumor cells [51]. In the present study, NK cell counts significantly decreased when exposed to hypobaric hypoxia training but was within the normal range of 7.2%–34.5%. B cells are a small lymphocyte subtype that is responsible for the humoral immunity component of the adaptive immune system by secreting antibodies [52]. Examples of autoimmune diseases, defined as diseases that are associated with B cell activity, include scleroderma, multiple sclerosis, systemic lupus erythematosus, type 1 diabetes, post-infectious irritable bowel syndrome, and rheumatoid arthritis [53]. T cells are a type of lymphocyte that develops in the thymus gland and play a central role in immune response. One of its functions is immune-mediated cell death, which is achieved in the following ways: CD8+ T cells, also known as “killer cells,” are cytotoxic, i.e., they can directly kill the virus-infected cells and cancer cells. A different population of T cells, the CD4+ T cells, functions as “helper cells” [54,55]. Different from CD8+ killer T cells, these CD4+ helper T cells indirectly kill cells identified as foreign: they determine if and how other parts of the immune system respond to a specific, perceived threat [54]. Regulatory T cells are yet another distinct population of these cells that provide the critical mechanism of tolerance, whereby immune cells are able to distinguish invading cells from “self,” thus preventing immune cells from inappropriately mounting a response against one’s cells [55]. In this study, NK cell counts significantly decreased when exposed to hypobaric hypoxia training but was within the normal range of 7.2%–34.5%. B cell and T cell counts significantly increased within the normal ranges (B cell, 6.2%–22.7%; T cell, 53.8%–81.8%) after the 2-week exercise training in hypobaric hypoxic conditions. This increase in the B cell and T cell counts is an important indicator of improvement in the adaptive immune system post-training. However, it was difficult to elucidate the improvement in immune function after hypoxic training performed in our study. Further studies assessing the mechanisms related to hypoxia training and immune function should be conducted.

In summary, 2-week exercise training in hypobaric hypoxic conditions for South Korean national cycling athletes with disabilities in our study had an unclear effect on immune function, but it did not affect their health function.

## 5. Limitations

The subjects of our study were the Korean national cyclists with disabilities and the sample size was limited. All six Korean national cyclists with disabilities participated in the study. Therefore, the present study was necessarily performed as a single trial pilot study. This is an obvious limitation of this study. Also, the study did not control extra variables, such as athletes’ diet, exercise, and supplement intake.

## 6. Conclusions and Suggestions

The present study revealed that 2-week exercise training in hypobaric hypoxic conditions has the potential to improve exercise performance in Korean national cycling athletes with disabilities. However, the effects of this training on immune function were unclear. To examine the effect of short-term hypoxic training on immune function in Korean national cycling athletes with disabilities, further studies and additional consultations, time, equipment, and research expenses are warranted to analyze additional associated variables. Subsequent studies should be able to derive clear study results from a control-experimental design based on sufficient sample size.

## Figures and Tables

**Figure 1 ijerph-17-00861-f001:**
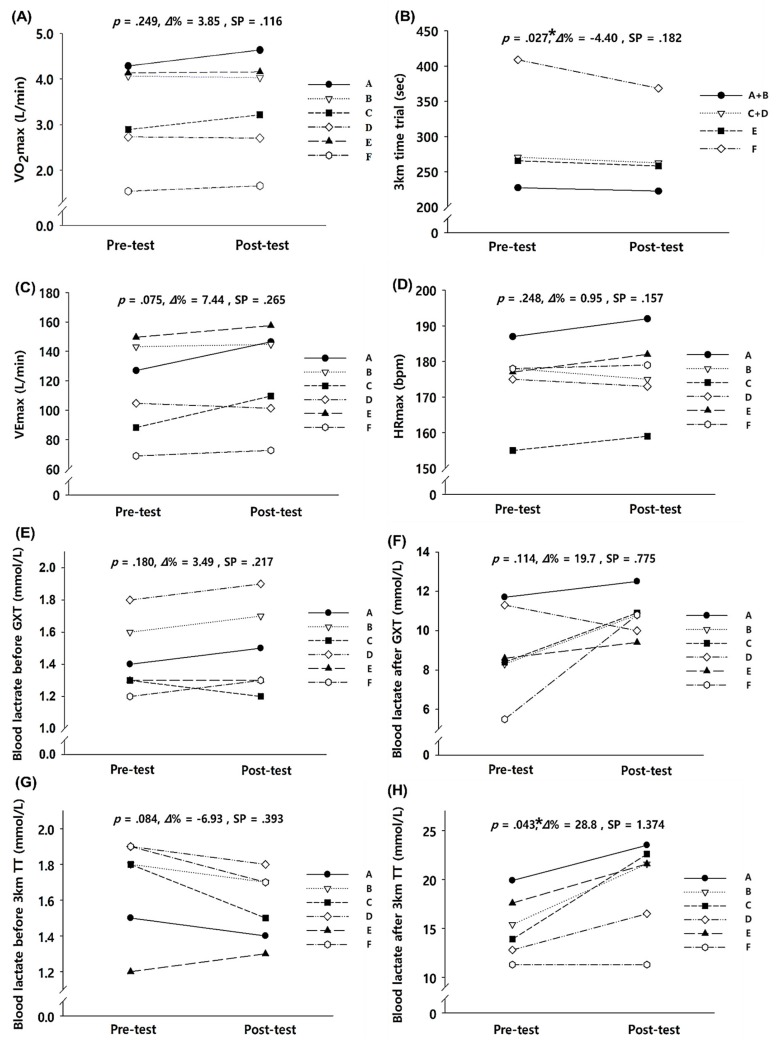
Exercise performance parameters before and after the 2-week exercise training in hypobaric hypoxia: (**A**) change in VO_2_max, (**B**) change in 3-km TT, (**C**) change in VEmax, (**D**) change in HRmax, (**E**) change in blood lactate before GXT, (**F**) change in blood lactate after GXT, (**G**) change in blood lactate level before 3-km TT., and (H) change in blood lactate level after 3-km TT. VO_2_max = maximal oxygen uptake, VEmax = maximal minute ventilation, HRmax = maximal heart rate, GXT = graded exercise test, TT = time trial, *Δ*% = delta percentage, SP = statistical power. * Significant difference between pre- and post-tests, *p* < 0.05.

**Figure 2 ijerph-17-00861-f002:**
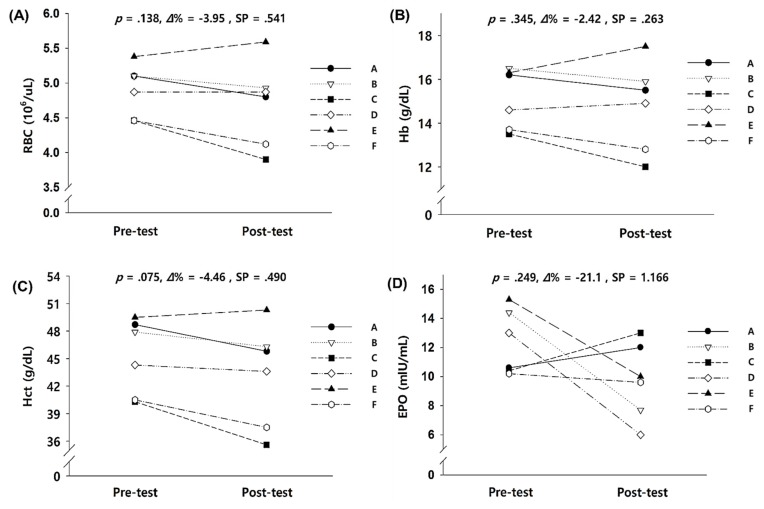
Oxygen-transporting capacity in the blood parameters before and after the 2-week exercise training in hypobaric hypoxia: (**A**) change in RBC, (**B**) change in Hb, (**C**) change in Hct, and (**D**) change in EPO. RBC = red blood cell, Hb = hemoglobin, Hct = hematocrit, EPO = erythropoietin, *Δ*% = delta percentage, SP = statistical power. ^*^ Significant difference between pre- and post-tests, *p* < 0.05.

**Figure 3 ijerph-17-00861-f003:**
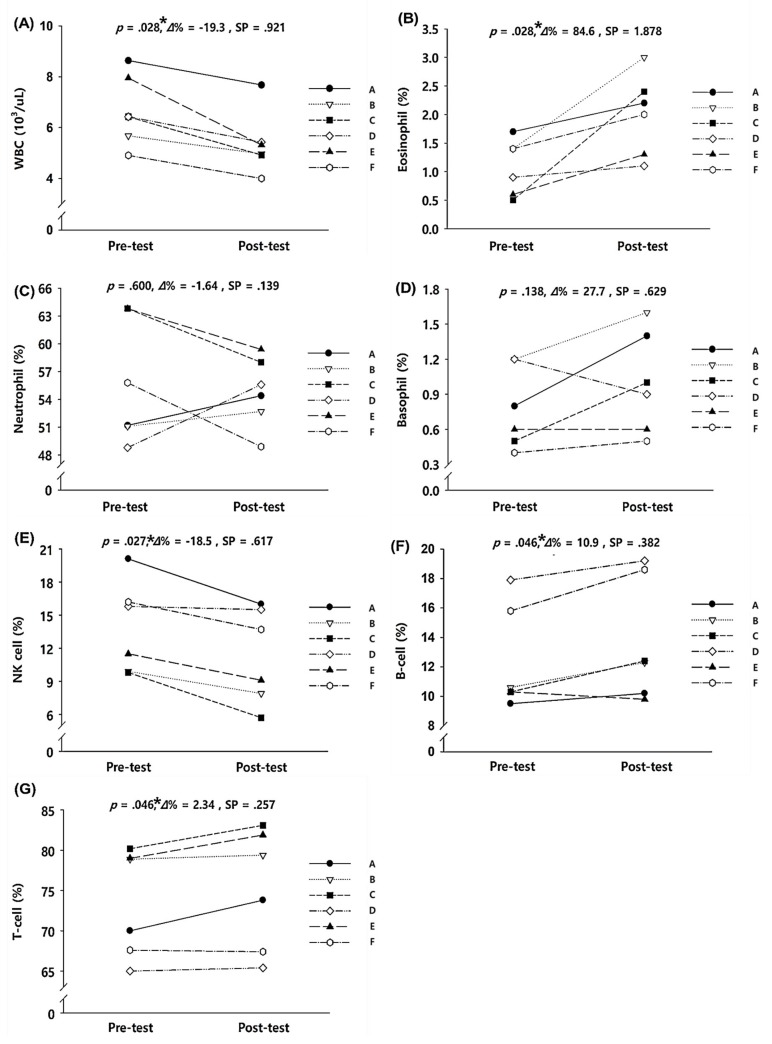
Immune function parameters before and after the 2-week exercise training in hypobaric hypoxia: (**A**) change in WBC, (**B**) change in eosinophil, (**C**) change in neutrophil, (**D**) change in basophil, (**E**) change in NK cell, (**F**) change in B cell, and (**G**) change in T cell. WBC = white blood cell, NK cell = natural killer cell. *Δ*% = delta percentage, SP = statistical power. ^*^ Significant difference between pre- and post-tests, *p* < 0.05.

**Table 1 ijerph-17-00861-t001:** Characteristics of the subjects.

Variables	Before Training	After Training	*p*-Value
Number (n)	6 (two blind athletes, one male and one female; two blind pilots, one male and one female; one cognitively-impaired male athlete; and one male athlete with a spinal cord disability)		
Environmental conditions (mmHg)	Week 1, 596 (simulated altitude: 2000 m); Week 2, 526 (simulated altitude: 3000 m).		
Age (years)	40.67 ± 8.36		
SBP (mmHg)	131.33 ± 11.67	137.17 ± 18.26	0.553
DBP (mmHg)	77.67 ± 7.69	69.37 ± 11.03	0.427
Weight (kg)	69.37 ± 11.03	68.70 ± 11.61	0.340
FFM (kg)	46.70 ± 12.23	49.27 ± 13.63	0.104
% Body fat	29.35 ± 10.38	28.57 ± 9.31	0.505

Values are expressed as mean ± standard deviation. SBP = systolic blood pressure, DBP = diastolic blood pressure, FFM = fat-free mass.

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
