# Peer review of "Effects of 2-Week Exercise Training in Hypobaric Hypoxic Conditions on Exercise Performance and Immune Function in Korean National Cycling Athletes with Disabilities: A Case Report"

_ijerph, 2020, doi:10.3390/ijerph17030861_

Round 1

Reviewer 1 Report

While this manuscript is interesting it has major methodological flaws. There is no justification of why to V02maxes were performed on a cycle ergometer but, then training was performed on a treadmill? With this said, the authors cannot compare changes in heart rate, v02max, VE because the modes were different. This is not a good designed study. Please report effect sizes and a power analysis is needed for the sample size. Why was 2 weeks chosen for training? For physiological changes to occur its takes around 4 weeks.

Introduction:

A better introduction on immune function is needed.

Methods:

143: Was the protocol for the v02max a typical proctol used for individuals with disabilities?

136: Was Blood pressure measured in duplicate?

Line 140: Please detail the body composition methods for DEXA.

144- 148. This is a run-on sentence.

149-151: The way this is worded it reads as you used a % of v02max variables. Please clarify this. Also, how were the metabolic variables analyzed (i.e. 15 second averaging)? What criteria was used to

Line 152: Was blood lactate measured in duplicate? Where was blood lactate measured from (ear or finger tip)?

Line 152: Was the 3 km time trial completed on the same day? Why was lactate higher during the time trial and not the v02max?

Results:

Tables 1 and 2: Please indicate what the Triangle means in the and put it in the abbreviation below.

Discussion:

Please restate the purpose in the discussion. The authors need to indicate future research questions. They need to note the sample size length of training as a limitation.

Author Response

Point-by-Point Responses to the Reviewers

We thank the reviewers for their guidance for further improving our revised manuscript (IJERPH-715305) entitled “Effects of a 2-Week Exercise Training in Hypobaric Hypoxic Condition on Exercise Performance and Immune Function in Korean National Cycling Athletes with Disabilities: A Pilot Study”. As described below, we have responded to all the comments brought up by the reviewers and incorporated all the changes suggested by the reviewer. Please note that the subjects of this study are the Korean national cyclists with disabilities and the number of subjects is limited. Currently, there are a total of six Korean national cyclists with disabilities and all athletes participated in this study. Therefore, this study was forced to proceed with a single trial pilot study. Please consider the limitation of this study. And we changed the article type from article to case report.

Reviewer 1.

Review Report Form

Comments and Suggestions for Authors

While this manuscript is interesting it has major methodological flaws. There is no justification of why to V02maxes were performed on a cycle ergometer but, then training was performed on a treadmill?

: Thanks for the good information and comments. This is our mistake. The athletes performed exercise training in hypoxia composed of continuous and interval training using a cycle as Korean national cyclists with disabilities. No treadmill training was done in this study. The contents of the manuscript have been revised.

With this said, the authors cannot compare changes in heart rate, v02max, VE because the modes were different.

: Although the athletes have different types of disabilities, there is no problem comparing their performance with hypoxic training. Only one study conducted hypoxic training for national cyclists with different disabilities, and this study also compared VO2max, HRmax, and all out time during GXY before and after training session [14]. After hypoxic training, increasing of VEmax via HVR and improving lactate tolerance are important adaptation for enhancing exercise performance.

This is not a good designed study. Please report effect sizes and a power analysis is needed for the sample size. Why was 2 weeks chosen for training? For physiological changes to occur its takes around 4 weeks.

: As mentioned above, there are currently Korean national disables cyclists with a total of six, all athletes participated in this study. That is, in our study, because the subject is special and there is a lack of number, the subject cannot be selected by calculating the sample size. Therefore, this study was forced to proceed with a single trial pilot study. Please consider the limitation of this study. The limitations of this study are described in addition to the manuscript and we changed the article type from article to case report.

Also, Korean national disables cycle team players usually have a schedule to compete every four weeks. So, training period is generally two to three weeks. It is common to have a training period of 4 to 6 weeks to improve athletic performance by exercise training in normoxia. However, exercise training in hypoxia vs normoxia has been reported to induce non-hemologic changes faster, thus improving exercise performance over short periods of time [4,8,14,21]. In addition, it is impossible to improve the national athlete's exercise performance with two weeks of exercise training at normoxia. In other words, it is very important to examine the effectiveness of hypoxic training that more effective than normoxic exercise for a short period on exercise performance. Thus, we set the hypoxic training period to two weeks.

Introduction:

A better introduction on immune function is needed.

: As mentioned in the introduction section, there is little research on changes in immune function following exposure to hypoxic conditions, and studies that examined the changes in immune function following exercise training in hypoxic condition in national athletes with disabilities have not been conducted. However, as reviewer 1 commented, we have further described the hypoxic condition and immune function in the introduction section.

Methods:

143: Was the protocol for the v02max a typical proctol used for individuals with disabilities?

: The GXT protocol used in the present study corresponds to the protocol generally used according to the disease when measuring vo2max in Korean National Cycling Athletes with Disabilities.

136: Was Blood pressure measured in duplicate?

: Blood pressure measurements were performed in duplicates and the average value was used as a result value. If the results of the first and second measurements were different, the measurement was repeated after 10 min of additional rest.

Line 140: Please detail the body composition methods for DEXA.

: The measurement details for DXA are as follows: Weight, fat-free mass, and percent body fat were analyzed as the body composition parameters using a dual-energy X-ray absorptiometry device (Primus, Osteosys, Seoul, Korea). To measure body composition using a dual-energy X-ray absorptiometry device, the subject was measured with the entire body in the center of the examination table, with both feet slightly rotated inward, and the shoulders and waist unmoved.

144- 148. This is a run-on sentence.

: This paragraph is two sentences and we changed to 2 sentences.

149-151: The way this is worded it reads as you used a % of v02max variables. Please clarify this. Also, how were the metabolic variables analyzed (i.e. 15 second averaging)? What criteria was used to

: In this study, respiratory gas data were measured every 10 seconds during GXT. The

VO2max was the highest VO2 obtained during the concluding period of the exercise test. The graded exercise test was completed when the following three items were satisfied: 1) VO2 plateau - no further increase in oxygen use per minute even with an increase in work performed, 2) HR within 10 beats of the age-predicted HRmax - this is the basis for using participants’ HRmax as a surrogate for the VO2max when designing personal training programs, and 3) plasma (blood) lactate concentrations of >7 mmol/L.

Also, we set the unit of VO2max to mL/kg/min and also indicate the rate of change by hypoxic training.

Line 152: Was blood lactate measured in duplicate? Where was blood lactate measured from (ear or finger tip)?

: Blood lactate level was measured in duplicate by finger-tip method using an automated blood lactate analyzer (Lactate Pro 2, ARKRAY Inc., Kyoto, Japan).

Line 152: Was the 3 km time trial completed on the same day? Why was lactate higher during the time trial and not the v02max?

: Pre- and post-tests were conducted over three days (two testing day and one resting day). Details are as follows - The experimental design involved the following: 2-day pretest session, a 2-week session of hypobaric hypoxic training, and 2-day post-test session.

All South Korean national cycling athletes with disabilities participated in a 2-day pre- and post-test session. On the first testing day, blood pressure and body composition were measured, and blood samples were collected between 8:00 and 10:00 AM after a day of fasting for the analysis of blood variables related to the oxygen-transporting capacity and immune function at sea level. Subsequently, after the athletes were allowed to eat and rest for 4 hours, the graded exercise test (GXT) were performed at normoxic condition. On the second testing day, the 3-km time trial test was conducted at Gwangmyeong Speedom, an official cycling stadium at normoxia.

Also, it is difficult to clearly explain why blood lactate level after 3 km time trial is higher than after GXT in the present study. However, we believe that at 3 km time trial, the exercise was performed at relatively higher average intensity for a shorter time than GXT. High intensity workouts during short periods of time will increase blood lactate levels. In particular, immediately after time trial, high blood lactate level shows high tolerance to fatigue substances and has a high correlation with exercise performance.

Results:

Tables 1 and 2: Please indicate what the Triangle means in the and put it in the abbreviation below.

: Δ% means delta percentage. A relative delta compares the difference between two numbers, A and B, as a percentage of one of the numbers. The basic formula is A - B/A x100.

Discussion:

Please restate the purpose in the discussion. The authors need to indicate future research questions. They need to note the sample size length of training as a limitation.

: As Reviewer 1's comment, we mentioned the purpose again in the discussion section.

In addition, the following suggestions were added: To examine the effect of short-term hypoxic training on immune function in Korean national cycling athletes with disabilities, further studies and additional consultations, time, equipment, and research expenses are warranted to analyze additional associated variables. Subsequent studies should be able to derive clear study results from a control-experimental design based on sufficient sample size.

Finally, the following limitations were added: The subjects of our study are the Korean national cyclists with disabilities and the sample size is limited. All six Korean national cyclists with disabilities participated in the study. Therefore, the present study was necessarily performed as a single trial pilot study. This is an obvious limitation of this study

Reviewer 2 Report

The authors examined the impacts of a 2 week aerobic continuous exercise and sprint interval exercise training in hypobaric hypoxic condition on performance and immune function. This topic is very interesting and very valuable, but this research design has some big limitations as a pilot study, thereby not supporting conclusion. For example, the author has not examined in control condition, such as without hypoxia, without exercise training, or in normal people, etc. So, this reviewer can not understand the “effective in improving the exercise performance” meaning (compared with what?).

Abstract

・Please re-consider about L18-L22 and L30-33, according to above and below comments.

Introduction

・Introduction is too long, whereas some information is not important for the rationale of this study. What is the main novelty and focusing purpose of this study? (e.g., Do the authors want to examine the impacts of training in hypoxia in athletes as compared with what? Only exercise effect? only hypoxia effects? or compared to normal subjects?)

・L93-95; This study only examined whether a 2 week aerobic continuous exercise and sprint interval exercise training in hypobaric hypoxic condition improves performance and immune function at Pre-Post, because of no control condition. In other words, this study did not examine that IMPACTS, such as compared to condition without hypoxia or exercise. Please rephrase it.

Methods

・How many female athletes did the authors include in this study?

・Did the authors control the diets and supplements in athlete while experiment?

Results

・Please show the absolutely value of VO2 max because body muscle content/volume should be changed after training. In addition to this, please statistical analysis on the data of SBP, DBP, Weight, FFM, and %Body fat.

・Please consider about presentation of individual plot as Figures because this study included small sample size.

Discussion

・Please discuss about the muscle glycogen content. As mentioned above, the diets and supplementation while experiment is an important factor for lactate production as well as exercise training impacts (e.g., glycogen loading). For example, if the participants performed same exercise, blood lactate concentration should be changed by muscle glycogen depletion (e.g., Tsukamoto et al. Repeated high-intensity interval exercise shortens the positive effect on executive function during post-exercise recovery in healthy young males. Physiol Behav. 2016). That is, muscle glycogen level may affect lactate production. Is it possible that muscle glycogen is affected by this training, thereby improving performance?

・Please compare to previous study data, if there is (e.g., The impact of only exercise, only hypoxia, or in normal subjects on these parameters). It would be to lead clearly rationale of this study, thereby indicating a strong significance.

・Please including limitations, such as without control conditions (participants), small sample size, and so on.

Conclusion

・Please re-consider according to previous my comments. Particularly, please clear the meaning of "effective for the improvement".

Author Response

Point-by-Point Responses to the Reviewers

We thank the reviewers for their guidance for further improving our revised manuscript (IJERPH-715305) entitled “Effects of a 2-Week Exercise Training in Hypobaric Hypoxic Condition on Exercise Performance and Immune Function in Korean National Cycling Athletes with Disabilities: A Pilot Study”. As described below, we have responded to all the comments brought up by the reviewers and incorporated all the changes suggested by the reviewer. Please note that the subjects of this study are the Korean national cyclists with disabilities and the number of subjects is limited. Currently, there are a total of six Korean national cyclists with disabilities and all athletes participated in this study. Therefore, this study was forced to proceed with a single trial pilot study. Please consider the limitation of this study. And we changed the article type from article to case report.

Reviewer 2.

Comments and Suggestions for Authors

The authors examined the impacts of a 2 week aerobic continuous exercise and sprint interval exercise training in hypobaric hypoxic condition on performance and immune function. This topic is very interesting and very valuable, but this research design has some big limitations as a pilot study, thereby not supporting conclusion. For example, the author has not examined in control condition, such as without hypoxia, without exercise training, or in normal people, etc. So, this reviewer can not understand the “effective in improving the exercise performance” meaning (compared with what?).

: Thank you for your kind comment in reviewer 2. As a reviewer 2 comments, there is a major limitation in the absence of exercise in normoxia as a control group in the present study. As mentioned above, there are currently Korean national disables cyclists with a total of six, all athletes participated in our study. That is, in the present study, because the subject is special and there is a lack of number, the subject cannot be selected by calculating the sample size. Therefore, this study was forced to proceed with a single trial pilot study. Please consider the limitation of this study. The limitations of this study are described in addition to the manuscript and we changed the article type from article to case report.

And, it is common to have a training period of 4 to 6 weeks to improve athletic performance by exercise training in normoxia. However, exercise training in hypoxia vs normoxia has been reported to induce non-hematologic changes faster, thus improving exercise performance over short periods of time [4,8,14,21]. Although there is no control group in our study, we consider that it is has an important meaning for elite athletes with disabilities to enhance their exercise performance by 2-week exercise training in hypoxic condition. We believe that the conclusion in our study is reasonable.

Abstract

・Please re-consider about L18-L22 and L30-33, according to above and below comments.

: Although there is no control group in our study, we consider that it is has an important meaning for elite athletes with disabilities to enhance their exercise performance by 2-week exercise training in hypoxic condition. In general, normoxic training for two weeks does not improve the performance of elite athletes. However, exercise training in hypoxia vs normoxia has been reported to induce non-hematologic changes faster, thus improving exercise performance over short periods of time [4,8,14,21]. In our study, the enhancement of exercise performance by two weeks of hypoxic training can be explained as having a very important physiological meaning. Especially, Korean national disables cycle team players usually have a schedule to compete every four weeks. So, training period is generally two to three weeks. Considering this situation, the enhancement of exercise performance by two weeks of exercise training in a hypobaric hypoxic condition can be described as a very new and meaningful result.

Introduction

・Introduction is too long, whereas some information is not important for the rationale of this study. What is the main novelty and focusing purpose of this study?

(e.g., Do the authors want to examine the impacts of training in hypoxia in athletes as compared with what? Only exercise effect? only hypoxia effects? or compared to normal subjects?)

: The present study evaluated the effects of exercise training in hypobaric hypoxic condition on exercise performance and immune function of Korean national cycling athletes with disabilities without control group (exercise training in normoxic condition).

Exercise training in hypoxic condition is known to enhance exercise performance in a short period more efficiently than exercise in normoxic condition. Exposure to the hypoxic condition has been reported to result in a change in immune function based on various physiological, metabolic, and neuroendocrine system changes. However, the study on changes of immune function following exercise training in the hypoxic condition is very insufficient. Various hypoxic training regimes are commonly used to enhance athletic performance in normoxic condition based on hematological and non-hematological changes, it is very important to examine the effects on immune function in terms of health and conditioning. Also, the World Anti-Doping Agency is concerned that various hypoxic training regimes can have a potentially negative effect on health. In other words, it is a very important task for elite athletes to examine how the exercise training in hypoxic condition affects immune function, and to secure the efficacy and stability of the hypoxic training. However, studies that examined the changes in immune function following exercise training in hypoxic condition in national athletes with disabilities have not been conducted. Particularly, it is important to examine the effects of exercise training in hypoxic condition on immune function in disabled individuals, who are more physiologically sensitive to immunological responses than nondisabled individuals.

Therefore, the present study aimed to examine the effects of a 2-week exercise training in hypobaric hypoxic condition corresponding to a simulated altitude of 2000–3000 m on exercise performance and immune function of Korean national cycling athletes with disabilities. We hypothesized that a 2-weeks exercise training in hypobaric hypoxic condition would enhance exercise performance, and it does not adversely affect immune function in Korean national cycling athletes with disabilities.

・L93-95; This study only examined whether a 2 week aerobic continuous exercise and sprint interval exercise training in hypobaric hypoxic condition improves performance and immune function at Pre-Post, because of no control condition. In other words, this study did not examine that IMPACTS, such as compared to condition without hypoxia or exercise. Please rephrase it.

: The present study evaluated the effects of exercise training in hypobaric hypoxic condition on exercise performance and immune function of Korean national cycling athletes with disabilities without control group (exercise training in normoxic condition). As mentioned earlier, although there is no control group in our study, we consider that it is has an important meaning for elite athletes with disabilities to enhance their exercise performance by 2-week exercise training in hypoxic condition. In general, normoxic training for two weeks does not improve the performance of elite athletes. However, exercise training in hypoxia vs normoxia has been reported to induce non-hematologic changes faster, thus improving exercise performance over short periods of time [4,8,14,21]. In our study, the enhancement of exercise performance by two weeks of hypoxic training can be explained as having a very important physiological meaning. Especially, Korean national disables cycle team players usually have a schedule to compete every four weeks. So, training period is generally two to three weeks. Considering this situation, the enhancement of exercise performance by two weeks of exercise training in a hypobaric hypoxic condition can be described as a very new and meaningful result.

Methods

・How many female athletes did the authors include in this study?

: The subjects of this study are as follows: Six South Korean national cycling athletes with disabilities included two blind athletes (one male and one female), two blind pilots (one male and one female), one cognitively impaired male athlete, and one male athlete with a spinal cord disability.

・Did the authors control the diets and supplements in athlete while experiment?

 : The study did not control extra variables, such as athletes' diet, exercise, and supplement intake. This is a limitation of this study.

Results

・Please show the absolutely value of VO2max because body muscle content/volume should be changed after training. In addition to this, please statistical analysis on the data of SBP, DBP, Weight, FFM, and %Body fat.

: As shown in Table 1, no significant change in body composition parameters (weight, FFM, and %body fat) in this study. In general, athletes do not show any changes in body composition after two to six weeks of endurance continuous and interval training. Rather, changes in body composition due to training are a sign of poor conditioning management and poor quality of training.

In addition, since there was no change in body composition parameters, it is appropriate to express VO2max as a relative value of weight as an exercise performance index.

Statistical analysis on the data of SBP, DBP, Weight, FFM, and% Body fat were added.

・Please consider about presentation of individual plot as Figures because this study included small sample size.

: In the present study, the sample size was small, but the number of dependent variables was large, and the table was used to accurately represent the changed values and ratios due to hypoxic training. In other words, we determined that representing data in a table would help readers better understand the content. If reviewer 2 strongly insists that the data should be represented using figures, we will change the table to figure in the next session.

Discussion

・Please discuss about the muscle glycogen content. As mentioned above, the diets and supplementation while experiment is an important factor for lactate production as well as exercise training impacts (e.g., glycogen loading). For example, if the participants performed same exercise, blood lactate concentration should be changed by muscle glycogen depletion (e.g., Tsukamoto et al. Repeated high-intensity interval exercise shortens the positive effect on executive function during post-exercise recovery in healthy young males. Physiol Behav. 2016). That is, muscle glycogen level may affect lactate production. Is it possible that muscle glycogen is affected by this training, thereby improving performance?

: Thank you for your good comment in reviewer 2. As a reviewer 2 comment, the muscle glycogen content is a factor affected by blood lactate levels. And hypoxic training has been reported to may increase muscle glycogen content by activating glycogen metabolism [Katayama et al., 2004; Hamlin et al., 2010; Galvin et al., 2013; Kasai et al., 2017]. However, we unfortunately did not measure muscle glycogen content in athletes. Also, our study did not control extra variables, such as athletes' diet, exercise, and supplement intake. Therefore, our study is difficult to discuss the change of blood lactate level according to glycogen content. This can also be explained as a limitation of this study.

But as mentioned in the manuscript, hypoxic training in the LLTH method has been reported change the non-hematological factors include increased oxidative enzyme activity [22,23], increased amount and density of the mitochondria [24-27], increased energy-utilizing capacity and changes in substrate utilization [9,28,29], enhanced blood lactate level tolerance and acid-base balance in the muscles [4,30], improved blood rheological and hemodynamic functions [10,31], enhanced intracellular iron delivery capacity [32], increased autonomic nervous system balance [33,34], changes in various hormone secretions [9,35], and increases in various proteins associated with oxygen utilization [36,37]. The physiological changes in the abovementioned non-hematological factors had positive effects on the VO2max and 3-km time trial records after our 2-week exercise training in hypobaric hypoxic condition. We focused on enhanced blood lactate level tolerance among several non-hematological factors. As a result, blood lactate tolerance and exercise performance were enhanced by exercise training in hypoxic condition for 2 weeks. These results are consistent with previous studies, and we can explain that the present study has originality and practicality in improving exercise performance through a short 2-week hypoxic training.

<Reference>

Katayama, K., Sato, K., Matsuo, H., Ishida, K., Iwasaki, K. I., & Miyamura, M. (2004). Effect of intermittent hypoxia on oxygen uptake during submaximal exercise in endurance

athletes. European Journal of Applied Physiology, 92(1-2), 75-83.

Hamlin, M. J., Marshall, H. C., Hellemans, J., Ainslie, P. N., & Anglem, N. (2010). Effect of intermittent hypoxic training on 20 km time trial and 30 s anaerobic performance. Scandinavian Journal of Medicine & Science in Sports, 20(4), 651-661. Galvin, H. M., Cooke, K., Sumners, D. P., Mileva, K. N., & Bowtell, J. L. (2013). Repeated sprint training in normobaric hypoxia. British Journal of Sports Medicine, 47(Suppl 1), i74-i79. Kasai N, Kojima C, Sumi D., Takahashi H., Goto K., Suzuki Y. (2017). Impact of 5 Days of Sprint Training in Hypoxia on Performance and Muscle Energy Substances. Int J Sports Med, 38(13): 983-991.

・Please compare to previous study data, if there is (e.g., The impact of only exercise, only hypoxia, or in normal subjects on these parameters). It would be to lead clearly rationale of this study, thereby indicating a strong significance.

: We wrote in detail L265-L279 what reviewer 2 commented on. The contents are as follows: in the present study, considering that we observed higher blood lactate levels after the 2-week exercise training in hypobaric hypoxic condition immediately after the 3-km time trial test, it can be inferred that short-term hypoxic training improves blood lactate tolerance, defined as the blood lactate’s ability to resist substances causing fatigue. Although the subjects and training period in hypoxic condition are different, Park & Lim [38] was to determine the effectiveness of 6-week intermittent hypoxic training (IHT) at 3000 m hypobaric hypoxic condition vs intermittent normoxic training (INT) on aerobic and anaerobic exercise capacity in competitive swimmers. IHT group showed a significant increase in VO2max and blood lactate level after exercise test compared with INT group. They insisted that IHT method may be effective in improvement of exercise performance by enhancing blood lactate tolerance in competitive swimmers. Hence, we believe that the 3-km time trial records of South Korean national cycling athletes with disabilities improved due to increased tolerance to blood lactate levels, and these findings are consistent with the findings in general elite athletes [4,38]. This increased tolerance to fatigue substances is considered beneficial for athletes who must repeatedly participate in high-intensity exercise training and competition.

・Please including limitations, such as without control conditions (participants), small sample size, and so on.

: We described the limitations of the study as follows: The subjects of our study are the Korean national cyclists with disabilities and the sample size is limited. All six Korean national cyclists with disabilities participated in the study. Therefore, the present study was necessarily performed as a single trial pilot study. This is an obvious limitation of this study. Also, the study did not control extra variables, such as athletes' diet, exercise, and supplement intake.

Conclusion

・Please re-consider according to previous my comments. Particularly, please clear the meaning of "effective for the improvement".

: Although there is no control group in our study, we consider that it is has an important meaning for elite athletes with disabilities to enhance their exercise performance by 2-week exercise training in hypoxic condition. In general, normoxic training for two weeks does not improve the performance of elite athletes. However, exercise training in hypoxia vs normoxia has been reported to induce non-hematologic changes faster, thus improving exercise performance over short periods of time [4,8,14,21]. In our study, the enhancement of exercise performance by two weeks of hypoxic training can be explained as having a very important physiological meaning. Especially, Korean national disables cycle team players usually have a schedule to compete every four weeks. So, training period is generally two to three weeks. Considering this situation, the enhancement of exercise performance by two weeks of exercise training in a hypobaric hypoxic condition can be described as a very new and meaningful result.

Round 2

Reviewer 1 Report

The authors need to provide a power analysis for future studies since this is a case report. 

Author Response

Point-by-Point Responses to the Reviewers

We thank the reviewers for their guidance for further improving our revised manuscript (IJERPH-715305) entitled “Effects of a 2-Week Exercise Training in Hypobaric Hypoxic Condition on Exercise Performance and Immune Function in Korean National Cycling Athletes with Disabilities: A Pilot Study”. As described below, we have responded to all the comments brought up by the reviewers and incorporated all the changes suggested by the reviewer.

Reviewer 1.

Review Report Form

Comments and Suggestions for Authors

The authors need to provide a power analysis for future studies since this is a case report.

: Thank you for the good comment. As reviewer 1 comments, we presented statistical power for all results.

Reviewer 2 Report

This manuscript is improved now as a case report, but this reviewer has some minor concerns as below. Hopefully, the rationale, significance, and novelty of this study will be more clear after minor revising.

Abstract

Again, this reviewer cannot understand "effective (L30-31)" meaning. This study has only indicated that the training in hypoxia in athletes improves exercise performance. In other words, we are NOT sure that the training in hypoxia in athletes is EFFECTIVE as compared with only exercise or only hypoxia exposure or in normal healthy people, from results of this study. For example, we cannot deny that only exercise training can get better exercise performance than combination of exercise and hypoxia. If so, it may be possible that the training in hypoxia in athletes is not effective strategy. Please rephrase it.

Introduction

Introduction is still too long, while there is some information which is not important for the rationale of this study as a case report. For example, a part of paragraph 1 is not necessary for this story. This reviewer recommends to compress the introduction, in order to easy-to-read and clear rationale of this study.

Results

Even though body composition parameters were no changed in this study, the absolutely value of VO2max should be presented because this study include small sample size (i.e., A Type 2 error is frequently). As the authors mentioned, exercise training in ATHLETES should not be changed body composition. That's especially why the absolutely value of VO2max should be similar to response of relative value. As the authors known, it is easy to change the body mass, so this study results would be strong if the authors present the absolute value of VO2 max.

This reviewer strongly recommends to present a Figure including individual plot. Because this study has small sample size; namely we cannot deny that type 2 error indicate no significant differences. The individual plot Figures (or at least effect size) would be better than delta percentage, in order to imply the changes in parameters.

Discussion (Limitations)

The authors should explain why these are factors the limitations of this study. This reviewer recommends to move limitations section to before conclusion because the authors need to discuss about limitation in order to clarify this paper position.

Conclusion

As commented in the abstract, "effective" is not suitable conclusion. Please rephrase it.

Author Response

Point-by-Point Responses to the Reviewers

We thank the reviewers for their guidance for further improving our revised manuscript (IJERPH-715305) entitled “Effects of a 2-Week Exercise Training in Hypobaric Hypoxic Condition on Exercise Performance and Immune Function in Korean National Cycling Athletes with Disabilities: A Pilot Study”. As described below, we have responded to all the comments brought up by the reviewers and incorporated all the changes suggested by the reviewer.

Reviewer 2.

Comments and Suggestions for Authors

Abstract

Again, this reviewer cannot understand "effective (L30-31)" meaning. This study has only indicated that the training in hypoxia in athletes improves exercise performance. In other words, we are NOT sure that the training in hypoxia in athletes is EFFECTIVE as compared with only exercise or only hypoxia exposure or in normal healthy people, from results of this study. For example, we cannot deny that only exercise training can get better exercise performance than combination of exercise and hypoxia. If so, it may be possible that the training in hypoxia in athletes is not effective strategy. Please rephrase it.

: Thank you for the good comment. We also agree with reviewer 2. The contents of L30-L31 have been revised as follows: These results indicate that our 2-week hypoxic training showed potential to improve exercise performance in Korean national disabled athletes. However, the effects of our hypoxic training method on the immune function remained unclear.

Introduction

Introduction is still too long, while there is some information which is not important for the rationale of this study as a case report. For example, a part of paragraph 1 is not necessary for this story. This reviewer recommends to compress the introduction, in order to easy-to-read and clear rationale of this study.

: We wrote the first paragraph to emphasize the need for efficient hypoxic training for disabled national team players. However, as the reviewer 2 comments, a part of paragraph 1 is not important for the purpose and hypothesis of the study. We deleted the first paragraph and briefly summarized the introduction section.

Results

Even though body composition parameters were no changed in this study, the absolutely value of VO2max should be presented because this study include small sample size (i.e., A Type 2 error is frequently). As the authors mentioned, exercise training in ATHLETES should not be changed body composition. That's especially why the absolutely value of VO2max should be similar to response of relative value. As the authors known, it is easy to change the body mass, so this study results would be strong if the authors present the absolute value of VO2 max.

: As the reviewer 2 comments, we changed the relative value (ml/kg/min) of VO2max to absolute value (L/min) in figure.

This reviewer strongly recommends to present a Figure including individual plot. Because this study has small sample size; namely we cannot deny that type 2 error indicate no significant differences. The individual plot Figures (or at least effect size) would be better than delta percentage, in order to imply the changes in parameters.

: As the reviewer 2 comments, we changed the Table 2-4 to individual plot Figure 1-3.

Discussion (Limitations)

The authors should explain why these are factors the limitations of this study. This reviewer recommends to move limitations section to before conclusion because the authors need to discuss about limitation in order to clarify this paper position.

: Thank you for the good comment. We move limitations section before conclusions and suggestions section.

Conclusion

As commented in the abstract, "effective" is not suitable conclusion. Please rephrase it.

: As mentioned earlier, we rephrased the sentence as follows: The present study revealed that a 2-week exercise training in hypobaric hypoxic condition showed potential to improve exercise performance (VO2max and 3-km time trial records) in Korean national cycling athletes with disabilities
